# Nonlinear topological symmetry protection in a dissipative system

Stéphane Coen [1,2] ✉, Bruno Garbin[1,2,7], Gang Xu [1,2,8], Liam Quinn[1,2], Nathan Goldman[3,4], Gian-Luca Oppo [5], Miro Erkintalo [1,2], Stuart G. Murdoch [1,2] & Julien Fatome[1,2,6]

We investigate experimentally and theoretically a system ruled by an intricate interplay between topology, nonlinearity, and spontaneous symmetry breaking. The experiment is based on a two-mode coherently-driven optical resonator where photons interact through the Kerr nonlinearity. In presence of a phase defect, the modal structure acquires a synthetic Möbius topology enabling the realization of spontaneous symmetry breaking in inherently bias-free conditions without fine tuning of parameters. Rigorous statistical tests confirm the robustness of the underlying symmetry protection, which manifests itself by a periodic alternation of the modes reminiscent of period-doubling. This dynamic also confers long term stability to various localized structures including domain walls, solitons, and breathers. Our findings are supported by an effective Hamiltonian model and have relevance to other systems of interacting bosons and to the Floquet engineering of quantum matter. They could also be beneficial to the implementation of coherent Ising machines.

Topology, nonlinearity, and spontaneous symmetry breaking are key concepts of modern physics. Topology is concerned with the study of properties that are invariant under continuous deformations, hence inherently robust against fluctuations and disorder[1,2]. Nonlinearity is the source of complexity, underpinning attractors, chaos, self-organization, and self-localization (solitons)[3,4]. Spontaneous symmetry breaking (SSB) underlies much of the diversity observed in Nature[5–7], from ferromagnetism[8] to embryo development[9]. Here, we report on an experiment in which these three concepts are fundamentally intertwined. Nonlinearity and SSB combine to endow our system with a topological invariant; in return, that invariance protects the underlying symmetry, enabling a robust realization of SSB that is immune to perturbations.

The discovery of topological states of matter in solid-state materials has revealed the central role of topology in the classification of quantum matter[1,10]. Besides, recent progress in designing and controlling synthetic lattice systems has allowed for the exploration of topological properties in a broad class of physical contexts[11,12], including ultracold gases[11], photonic crystals[12,13], mechanics[14], and systems with synthetic dimensions[13,15]. Topological robustness has potential wide-ranging applications, from fault-tolerant quantum computers[16,17] to lasers capable of single-mode operation at high power[18]. While topological band structures concern the properties of single-particle Bloch states, interesting behaviors were observed by combining these with classical nonlinearities[19]. This includes, non-exhaustively, edge solitons, nonlinearity-induced topological phase

[1]Physics Department, The University of Auckland, Private Bag 92019, Auckland 1142, New Zealand. [2]The Dodd-Walls Centre for Photonic and Quantum Technologies, Dunedin, New Zealand. [3]Center for Nonlinear Phenomena and Complex Systems, Université Libre de Bruxelles, CP 231, B-1050 Brussels, Belgium. [4]Laboratoire Kastler Brossel, Collège de France, CNRS, ENS-Université PSL, Sorbonne Université, 11 Place Marcelin Berthelot, 75005 Paris, France. [5]SUPA and Department of Physics, University of Strathclyde, Glasgow G4 0NG, Scotland. [6]Laboratoire Interdisciplinaire Carnot de Bourgogne (ICB), UMR 6303 CNRS, Université de Bourgogne, 9 Avenue Alain Savary, BP 47870, F-21078 Dijon, France. [7]Present address: NcodiN SAS, 10 Boulevard Thomas Gobert, F-91120 Palaiseau, France. [8]Present address: School of Optical and Electronic Information, Huazhong University of Science and Technology, 1037 Luoyu Road, Wuhan, China. ✉e-mail: s.coen@auckland.ac.nz

transitions[20,21], novel symmetry-protected phases[22], or quantized Thouless pumping of solitons[23].

The results presented in this Article demonstrate a new facet of the potent bridging of topology and nonlinearity and how it can confer robustness to the realization of SSB. Upon SSB, a system bifurcates to states with a broken symmetry even though the equations of motion retain that symmetry[5–7]. Intense interest has been paid to this process in recent years, especially in optical microresonators where it has been studied, e.g., for optical memories[24], nonreciprocal propagation in integrated photonic circuits[25,26], dynamical control of laser directionality, chirality, and polarization[27,28], or sensors with divergent sensitivities[29,30]. Experiments on SSB have also been used for fundamental studies of domain walls[31], of the universal unfolding of the pitchfork bifurcation[32,33], or of the Bose–Hubbard dimer model[34]. These experiments most often rely on an exchange symmetry ($\mathbb{Z}_2$) between two states. Invariably, however, this symmetry is affected by manufacturing imperfections, experimental biases, or other non-idealities. As a consequence, instead of a spontaneous, random selection between the two states, one state is statistically favored over the other[33,35], hindering the experiments. Here, we show how the combination of nonlinearity and topology can solve that problem. Specifically, we report on the experimental realization of an optical resonator with a synthetic modal structure characterized by a Möbius topology protected by the Kerr nonlinearity. That protection leads to an exact exchange symmetry, which enables robust, bias-free SSB that does not require any fine-tuning of the system parameters. The generation of truly random binary sequences based on this platform confirms the robustness of the exchange symmetry. Additionally, we reveal a new class of topological symmetry-broken localized structures characterized by rapid periodic exchange of their symmetry. Our observations are supported by theory, which reveals, in particular, the key role played by degenerate parametric processes. This connects our work with some recent implementations of so-called coherent Ising machines[36,37] that could potentially benefit from our symmetry-protection scheme. These benefits could extend to experiments on Bose–Einstein condensates. Indeed, the Kerr nonlinearity stems from interaction processes among photons, in direct analogy with the nonlinearity inherent to Bose-Einstein condensates and described by the Gross-Pitaevskii equation[38]. We also present an effective Hamiltonian approach, similar to that used for the Floquet engineering of quantum matter[39–41], to explain aspects of the emergence of symmetry in our system. Finally, our work could have relevance to the symmetry

restoration techniques used in mean-field approaches of quantum many-body systems in nuclear, atomic, and molecular physics[42].

## Results

### Principle of symmetry protection

Our system consists of a passive nonlinear (Kerr) optical resonator coherently driven by a single-frequency field that presents two distinct polarization mode families. Hereafter $\psi_x$ and $\psi_y$ denote the modes' classical electric field (complex) amplitudes inside the resonator scaled such that $|\psi_x|^2$ and $|\psi_y|^2$ represent the number of photons in each mode. In this configuration, the nonlinearity is well known to induce a polarization SSB bifurcation contingent on the existence of an exchange symmetry between the two modes[43,44]. That symmetry is typically realized by being as close as possible to conditions where the modes are equally driven and equally detuned from the driving field frequency[25,26,33]. Here, we deviate from this paradigm. Specifically, we introduce a localized $\pi$-phase-shift defect between the two mode families of the resonator. As a result, their resonance frequencies are separated by half a free-spectral range (FSR). In these conditions, the exchange symmetry cannot be realized as described above: the single-frequency driving field can only be resonant with one of the modes, assumed here to be the $x$ mode; hence, only that mode is driven, and the roundtrip phase detunings differ by $\pi$. These different aspects are illustrated schematically in Fig. 1a.

Consider now an alternative description of this resonator in terms of the hybrid mode amplitudes $\psi_+$ and $\psi_-$ defined by the unitary transformation

$$\psi_+ = \frac{1}{\sqrt{2}}\left(\psi_x + i\psi_y\right), \psi_- = \frac{1}{\sqrt{2}}\left(\psi_x - i\psi_y\right). \tag{1}$$

Because of the $\pi$-phase defect, the amplitude of the $y$ mode flips its sign with respect to the driven mode at each roundtrip in the resonator, $\psi_y \rightarrow \psi_y e^{i\pi} = -\psi_y$. From the above relations, it is clear that this is associated with a roundtrip-to-roundtrip periodic swapping of the + and − hybrid mode amplitudes, $\psi_+ \rightleftharpoons \psi_-$ [right side in Fig. 1c]. The field must circulate two roundtrips in the resonator to be restored to its original state, revealing that the hybrid modes embody the topology of a Möbius strip[45–49]. In terms of those hybrid modes, the resonator shown in Fig. 1a is thus equivalent to that represented in Fig. 1b. Note that the Möbius topology only exists in the synthetic modal dimension rather than in real physical space[50]. A consequence of the single-sided,

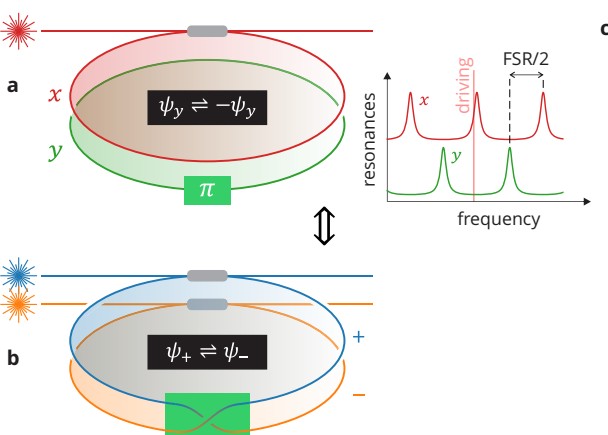
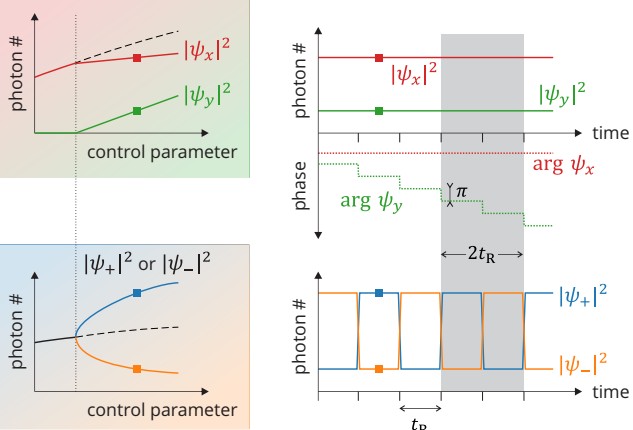

**Fig. 1 | Principle of operation. a** Schematic representation of a two-mode coherently driven Kerr resonator with a $\pi$-phase defect. The inset on the right shows the linear resonances of the two modes and how only one mode is driven (here, the $x$ mode). **b** Equivalent Möbius-like resonator described in terms of the + and − hybrid modes defined by Eqs. (1). **c** Parametric oscillation of the $y$-mode, $|\psi_y|^2 > 0$ (top left)

corresponds to an SSB bifurcation for the occupations of the hybrid modes, $|\psi_+|^2 \neq |\psi_-|^2$ (bottom left). In that regime (P2 SSB), the phase of the $y$-mode changes by $\pi$ at each resonator roundtrip (top right), corresponding to a periodic roundtrip-to-roundtrip swapping of the hybrid mode occupations (bottom right). $t_R$ is the roundtrip time of the resonator.

non-orientable nature of the Möbius topology is that the two hybrid mode amplitudes will necessarily accumulate the same changes over two resonator roundtrips. The resonator, therefore, exhibits an exchange symmetry, $\psi_+ \rightleftharpoons \psi_-$. That symmetry, coupled with the nonlinearity, enables SSB between the hybrid modes.

Here, it is important to realize that SSB in our system is fundamentally intertwined with the generation of the $y$ field. As can be seen from Eqs. (1), a symmetric state for the hybrid modes, $\psi_+ = \psi_-$, is necessarily associated with a $y$ field of zero amplitude. By contrast, a symmetry-broken hybrid state, $\psi_+ \neq \psi_-$, requires a non-zero $y$ field. By symmetry, the mirror state, $\psi_+ \rightleftharpoons \psi_-$, must also exist, corresponding to a $y$ field with a phase differing by $\pi$, $\psi_y \rightleftharpoons -\psi_y$. Keeping in mind that the $y$-field is undriven, it must be generated internally through nonlinear interactions past a certain threshold of detuning and/or driving strength. What appears as an SSB bifurcation for the populations $|\psi_+|^2$, $|\psi_-|^2$ of the + and − hybrid modes can, therefore, be seen as the threshold of parametric oscillation for the $y$ mode[51] [left side in Fig. 1c]. However, our system is not just a standard parametric oscillator. Above the bifurcation threshold, the underlying Möbius topology imparted by the $\pi$ phase defect forces the system to alternate between the two symmetry-broken states one roundtrip to the next and to display an overall periodicity of two roundtrips. This is consistent with the spectral components of the $y$-mode being generated near their corresponding resonance condition, i.e., shifted by ±FSR/2 from the driving field frequency. As a reminder of these specific characteristics, we will refer to this regime as P2 SSB. One can also note that, in the P2 SSB regime, the amplitudes of the two hybrid modes over one resonator roundtrip become indistinguishable from the amplitude of one of those modes over two subsequent roundtrips. An alternative interpretation is thus that SSB now occurs between subsequent resonator roundtrips, in essence breaking the time translation symmetry in a way reminiscent of period doubling instabilities[48,52–54].

## Theoretical analysis

The conclusions of the previous Section are based on the premise that the phase defect seen by the $y$ mode at each resonator roundtrip is exactly $\pi$. To understand how the Möbius topology and the associated symmetry can persist in the presence of deviations from this condition, we must examine some aspects in more detail. To that end, we introduce the Hamiltonian $\hat{H}_{NL}$ describing the Kerr-mediated interactions taking place along the resonator roundtrip,

$$\frac{1}{\hbar g}\hat{H}_{NL} = \frac{A}{2}\left(\hat{a}_x^\dagger\hat{a}_x^\dagger\hat{a}_x\hat{a}_x + \hat{a}_y^\dagger\hat{a}_y^\dagger\hat{a}_y\hat{a}_y\right) + B\left(\hat{a}_x^\dagger\hat{a}_y^\dagger\hat{a}_x\hat{a}_y\right) + \frac{C}{2}\left(\hat{a}_x^\dagger\hat{a}_x^\dagger\hat{a}_y\hat{a}_y + \hat{a}_y^\dagger\hat{a}_y^\dagger\hat{a}_x\hat{a}_x\right). \tag{2}$$

Here $g$ is the single photon-induced Kerr frequency shift of the resonator ($g > 0$)[55] while $\hat{a}_\sigma^\dagger$ (resp. $\hat{a}_\sigma$) represents creation (annihilation) operators for the mode $\sigma = x, y$. These operators satisfy bosonic commutation relations, $[\hat{a}_\sigma, \hat{a}_{\sigma'}^\dagger] = \delta_{\sigma,\sigma'}$. The three terms on the right-hand side of Eq. (2) describe, respectively, on-site (Bose−Hubbard) interactions (self-phase modulation), inter-site interactions (cross-phase modulation), and pair hopping (parametric four-photon mixing). The latter corresponds to two photons in the $x$ mode being converted to two photons in the $y$ mode (or vice versa); since the $y$-mode is undriven, this term is required as the source of $y$ photons. Without loss of generality, we will assume $A = 1$, $B = 2/3$, and $C = 1/3$. These are the values found in silica (which is the Kerr material used in our experimental demonstration presented below) for modes that are linearly polarized[56,57]. Generalization to other polarization states will be discussed in the Methods.

To understand specifically the evolution of the phase of the $y$ mode, we examine the equation of motion for $\hat{a}_y$, i.e.,

$d\hat{a}_y/dt = i[\hat{H}_{NL}, \hat{a}_y]/\hbar$. In the classical limit, $\hat{a}_y^\dagger \rightarrow \psi_y$, we get

$$\frac{d\psi_y}{dt} = ig\left[\left(A|\psi_y|^2 + B|\psi_x|^2\right)\psi_y + C\psi_y^*\psi_x^2\right], \tag{3}$$

where the right-hand side has the form of the interaction terms found in the Gross-Pitaevskii/nonlinear Schrödinger equation (* denotes complex conjugation). Separating amplitude and phase, $\psi_\sigma = |\psi_\sigma|e^{i\phi_\sigma}$, we find that the phase $\phi_y$ of the $y$ mode obeys

$$\frac{d\phi_y}{dt} = g\left[A|\psi_y|^2 + B|\psi_x|^2 + C|\psi_x|^2\cos\left(2\phi_y - 2\phi_x\right)\right]. \tag{4}$$

The three terms in the above equation represent phase shifts induced by the Kerr nonlinearity. Without loss of generality and for simplicity, we can restrict ourselves to high-Q resonators where all these terms can be assumed $\ll 1$. Indeed, nonlinear effects will only have an important role when close to resonance. In these conditions, we can use a single Euler step to approximate the integration of Eq. (4) over one resonator roundtrip time $t_R$. To obtain the total change in the phase of the $y$ mode from one roundtrip to the next, we still need to consider the role of the boundary conditions, which add a linear contribution due to the resonator detuning (see Methods for more detail), leading to

$$\phi_y^{(m+1)} - \phi_y^{(m)} = gt_R\left(A|\psi_y|^2 + B|\psi_x|^2\right) + gt_R C|\psi_x|^2\cos\left(2\phi_y^{(m)} - 2\phi_x^{(m)}\right)\underbrace{-\delta_0 + \pi + \delta_\pi}_{-\delta_y}. \tag{5}$$

Here $m$ is the roundtrip index, $\delta_0 = \delta_x$ is the roundtrip phase detuning of the driving field (modulo $2\pi$), also assumed $\ll 1$ ($\delta_0 > 0$ means red-detuned), and the corresponding detuning for the $y$ mode is defined through $\delta_x - \delta_y = \pi + \delta_\pi$, with $\delta_\pi$ representing deviations from an exact value of $\pi$. We can observe that all terms above are, in fact, $\ll \pi$, except $\pi$ itself (a similar argument for the $x$ mode would show that $\phi_x^{(m+1)} - \phi_x^{(m)} \ll 1$). Moreover, the pair hopping (parametric) contribution ($\propto C$), which is the only phase-sensitive term, is $\pi$ periodic in $\phi_y$. This is a well-known characteristic of degenerate parametric oscillators[58]. It follows from Eq. (5) that the only stable configuration is for the phase of the $y$ mode to step by exactly $\pi$, one roundtrip to the next. That solution acts as an attractor for the system, irrespective of the value of $\delta_\pi$. Effectively, the intracavity fields will self-adjust through the dissipative nonlinear dynamics to guarantee a roundtrip-to-roundtrip $y$ mode phase step of $\pi$. In this way, the Möbius topology of the resonator is intrinsically robust and protected by nonlinearity.

As a second step, we examine how the Möbius topology induces an exact exchange symmetry $\psi_+ \rightleftharpoons \psi_-$. To that end, we introduce an explicit asymmetry and show how it cancels out. Specifically, we consider a difference in detunings between the + and − hybrid modes. The Hamiltonian Eq. (2) can be expressed in terms of these modes with creation (annihilation) operators $\hat{a}_\pm^\dagger$ ($\hat{a}_\pm$); these are related to the corresponding $x, y$ operators by the same unitary transformations as Eqs. (1). We find

$$\frac{1}{\hbar g}\hat{H}_{NL} = \frac{A'}{2}\left(\hat{a}_+^\dagger\hat{a}_+^\dagger\hat{a}_+\hat{a}_+ + \hat{a}_-^\dagger\hat{a}_-^\dagger\hat{a}_-\hat{a}_-\right) + B'\left(\hat{a}_+^\dagger\hat{a}_+^\dagger\hat{a}_+\hat{a}_-\right) \tag{6}$$

where $A' = 2/3$ and $B' = 4/3$. For convenience, let us define a set of angular momentum (Schwinger) operators as

$$\hat{J}_x = \frac{\hbar}{2}\left(a_+^\dagger a_- + a_-^\dagger a_+\right), \quad \hat{J}_y = \frac{\hbar}{2i}\left(a_-^\dagger a_+ - a_+^\dagger a_-\right),$$
$$\hat{J}_z = \frac{\hbar}{2}\left(a_-^\dagger a_- - a_+^\dagger a_+\right), \quad \hat{N} = a_+^\dagger a_+ + a_-^\dagger a_-. \tag{7}$$

These operators satisfy commutation relations $[\hat{J}_\mu, \hat{J}_\nu] = i\hbar \epsilon_{\mu\nu\lambda} \hat{J}_\lambda$ similar to spins ($\epsilon_{\mu\nu\lambda}$ is the Levi–Civita symbol), while $\hat{N}$ counts the total number of photons. In this part of our analysis, we assume a lossless resonator (conservative limit) so $\hat{N}$ is constant. With these notations, a difference in angular frequency detuning $\delta\Omega$ between the two hybrid modes enters the Hamiltonian as $\delta\Omega \hat{J}_z$, so that the total Hamiltonian of the system reads

$$\begin{aligned} \hat{H}_{\text{asym}} &= \hat{H}_{\text{NL}} + \delta\Omega \hat{J}_z \\ &= \hbar g \left[ a \hat{J}_z^2/\hbar^2 + b \hat{N}^2 - c\hat{N} \right] + \delta\Omega \hat{J}_z \end{aligned} \tag{8}$$

where $a = A' - B'$, $b = (A' + B')/4$, and $c = A'/2$.

We now proceed by seeking an effective (Floquet) Hamiltonian $\hat{H}_{\text{eff}}$[39,59], which captures stroboscopically the dynamics of a full period of the Möbius configuration depicted in Fig. 1b, i.e., every two actual roundtrips of the resonator. The swapping of the two hybrid modes at each roundtrip is introduced through the discrete application of the operator $\hat{U}_{\text{swap}} = \hat{\sigma}_x = e^{i(\pi/2)(\hat{\sigma}_x - 1)}$ (with $\hat{\sigma}_x$ a Pauli matrix), leading to

$$e^{-i\frac{1}{\hbar}2t_R \hat{H}_{\text{eff}}} = \underbrace{\hat{U}_{\text{swap}} e^{-i\frac{1}{\hbar}t_R \hat{H}_{\text{asym}}} \hat{U}_{\text{swap}}}_{e^{-i\frac{1}{\hbar}t_R \hat{H}_1}} e^{-i\frac{1}{\hbar}t_R \hat{H}_{\text{asym}}}. \tag{9}$$

The first three factors on the right-hand side of this expression define $\hat{H}_1$ which can be obtained exactly as[41]

$$\hat{H}_1 = e^{i\frac{\pi}{2}\hat{\sigma}_x} \hat{H}_{\text{asym}} e^{-i\frac{\pi}{2}\hat{\sigma}_x}. \tag{10}$$

Noting that the swapping operation is a single-particle process, we can substitute $\hat{J}_x/\hbar$ for $\hat{\sigma}_x/2$ in the exponentials above and then use the Baker–Campbell–Hausdorff formula. Specifically, we have $e^{i\pi \hat{J}_x/\hbar} \hat{J}_z e^{-i\pi \hat{J}_x/\hbar} = -\hat{J}_z$ while all other terms of $\hat{H}_{\text{asym}}$ are left unchanged, leading to

$$\hat{H}_1 = \hat{H}_{\text{NL}} - \delta\Omega \hat{J}_z. \tag{11}$$

Comparing the expressions for $\hat{H}_{\text{asym}}$, Eq. (8), and $\hat{H}_1$, above, we observe that the sign of the asymmetric term is reversed. It is also clear that $\hat{H}_{\text{asym}}$ commutes with $\hat{H}_1$, which from Eq. (9) leads to $\hat{H}_{\text{eff}} = (\hat{H}_1 + \hat{H}_{\text{asym}})/2$. It follows that the asymmetric term cancels out exactly over two resonator roundtrips, and we have $\hat{H}_{\text{eff}} = \hat{H}_{\text{NL}}$. Our analysis therefore shows that the effective Hamiltonian, which describes propagation every two resonator roundtrips, including two swaps of the hybrid modes, only contains the original–symmetric–nonlinear interactions; it is immune to fluctuations and asymmetries in detunings as foreseen in our qualitative description of the previous Section. This confirms the existence of an exact exchange symmetry for the hybrid modes, $\psi_+ \rightleftharpoons \psi_-$, and the topological robustness of the P2 SSB regime.

We are now in a position to derive mean-field equations of motion for the classical fields $\psi_+$, $\psi_-$ based on the effective Hamiltonian $\hat{H}_{\text{eff}}$. We have already established that the dissipation guarantees an exact overall roundtrip phase jump of $\pi$ for the $y$ mode, so we do not need to consider the influence of $\delta_\pi$, the deviation from an exact $\pi$ phase defect, on the swapping operator $\hat{U}_{\text{swap}}$. Still, $\delta_\pi$ contributes to the linear detunings through the relation $\delta_0 - \delta_y = \pi + \delta_\pi$. This effect can be taken into account by the following roundtrip phase detuning operator,

$$\hbar \left[ \delta_0(\hat{a}_x^\dagger \hat{a}_x) + (\delta_0 - \delta_\pi)(\hat{a}_y^\dagger \hat{a}_y) \right] = \hbar \left( \delta_0 - \frac{\delta_\pi}{2} \right) \hat{N} + \delta_\pi \hat{J}_x. \tag{12}$$

Note that, for the $y$ mode, we have discarded the $\pi$ phase-shift that is already incorporated in $\hat{H}_{\text{eff}}$. The contribution of the detunings can be

distributed along the resonator roundtrip and combined with the effective Floquet Hamiltonian, leading to, in the classical limit,

$$\begin{aligned} t_R \frac{d\psi_+}{dt} = &\left[ -\frac{t_R \Delta\omega}{2} + i g t_R (A'|\psi_+|^2 + B'|\psi_-|^2) - i\left( \delta_0 - \frac{\delta_\pi}{2} \right) \right] \psi_+ \\ &- i\frac{\delta_\pi}{2} \psi_- + \sqrt{\theta} S, \end{aligned} \tag{13}$$

$$\begin{aligned} t_R \frac{d\psi_-}{dt} = &\left[ -\frac{t_R \Delta\omega}{2} + i g t_R (A'|\psi_-|^2 + B'|\psi_+|^2) - i\left( \delta_0 - \frac{\delta_\pi}{2} \right) \right] \psi_- \\ &- i\frac{\delta_\pi}{2} \psi_+ + \sqrt{\theta} S. \end{aligned} \tag{14}$$

To these equations, we have added phenomenologically losses (dissipation) associated with the linewidth $\Delta\omega$ of the resonances and driving, with strength $S$ and external coupling fraction $\theta$. In the conservative limit applied above, these terms are not included directly. An alternative classical derivation of the mean-field equations (13)–(14), taking into account these dissipative effects, is presented in the Methods for completeness. That derivation confirms, in particular, that the Möbius topology leads, as for the detunings, to a robust symmetrization of the driving. Note that the symmetry of the driving can also be interpreted as stemming from the absence of driving of the $y$ mode; under these conditions, Eqs. (1) indeed entails $S_+ = S_- = S = S_x/\sqrt{2}$, where $S_x$ is the driving strength of the $x$ mode only. Additionally, we can observe that a deviation $\delta_\pi$ from an exact $\pi$ phase defect simply causes a common shift in detuning as well as some linear coupling between the hybrid modes, none of which alters the exchange symmetry of the equations. Conversely, we can interpret the equality of the detunings as stemming from the absence of linear mode coupling between the $x$ and $y$ modes due to the associated fields having distinct spectral components.

Apart from the absence of kinetic terms, the role of which will be discussed below, Eqs. (13)–(14) have the form of coupled, driven, damped, Gross–Pitaevskii/nonlinear Schrödinger equations. These equations are well known to exhibit SSB, provided the detunings and the driving strengths of the two modes are identical[25,26,30,33,43,44]. Contrary to previous implementations, in our system, and thanks to the presence of the $\pi$ phase defect, this condition is automatically and robustly satisfied. Finally, while the equations above describe the stroboscopic mean-field evolution of $\psi_+$ and $\psi_-$ two resonator roundtrips at a time, it is important to keep in mind that the amplitudes of the two fields swap at each resonator roundtrip due to the underlying Möbius topology.

## Experimental observation of P2 SSB

To demonstrate experimentally the topologically-protected P2 SSB regime, we have used, as modes $x$ and $y$, the two polarization modes of a ring resonator constructed from single-transverse mode silica optical fiber. This platform enables straightforward implementation of both the $\pi$ phase-shift defect and the projection of the $x$ and $y$ modes onto the $+$ and $-$ hybrid modes through manipulations of the polarization state of light with optical fiber polarization controllers (FPCs)[60]. Note that for $x$ and $y$ modes that are orthogonally linearly polarized, the $+$ and $-$ hybrid modes as defined by Eqs. (1) correspond to circular polarization states of opposite handedness.

A schematic of the experimental setup is depicted in Fig. 2. As we used slightly different resonators with different specifications for different measurements (in part to demonstrate the universality of the methodology), we list here the parameters corresponding to the results presented in this Section. Variations between setups will be highlighted as appropriate. To facilitate comparisons, experimental results will be presented in terms of normalized parameters. Specifically, we will be referring to the normalized roundtrip phase detuning

$\Delta = \delta_0/\alpha$ and the normalized driving power $X = g t_R \theta S_x^2/\alpha^3$, where $\alpha = t_R \Delta\omega/2$ is the dissipation rate of Eqs. (13)–(14).

Our first resonator is built around a fiber coupler (beam-splitter) that recirculates 90% of the intracavity light. This coupler is also used for the injection of the coherent driving field ($\theta = 0.1$). Another 1% tap coupler extracts a small fraction of the intracavity field for analysis. Both couplers are made up of standard SMF28 (Corning) optical fiber. The rest of the resonator is made up of a highly nonlinear, low-birefringence spun fiber (iXblue Photonics)[61]. At the 1550 nm driving wavelength, that fiber exhibits normal group-velocity dispersion (corresponding to repulsive interactions or a positive effective mass), which has been selected to avoid scalar parametric instabilities[57]. Overall, the resonator is 12 m long, corresponding to a roundtrip time $t_R$ of 57 ns (FSR = 17.54 MHz), and has a resonance linewidth $\Delta\omega/(2\pi) = 650$ kHz. The resonator is driven by a 1550 nm-wavelength continuous-wave laser (NKT Photonics) with an ultra-narrow linewidth < 1 kHz, ensuring coherent driving. The laser frequency can be varied with a piezoelectric transducer, which provides the ability to scan the detuning. Alternatively, we can use the transducer to lock the detuning at a set value via a feedback loop using the technique discussed in ref. 62. The driving laser is intensity-modulated with a Mach–Zehnder amplitude modulator driven by a pulse pattern generator to generate flat-top 1.1 ns-long pulses with a period matching the resonator roundtrip time $t_R$[63]. Before injection into the cavity, the driving pulses are amplified with an erbium-doped fiber amplifier followed by an optical bandpass filter, which reduces amplified spontaneous emission noise.

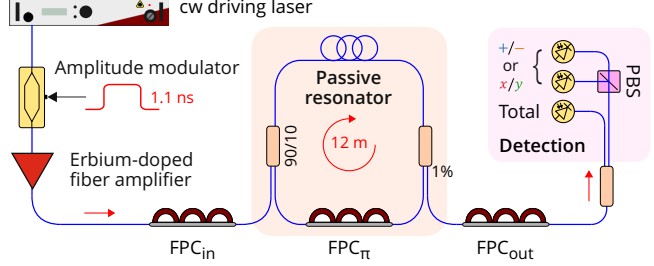

**Fig. 2 | Schematic of the experimental setup.** FPC optical fiber polarization controller, PBS polarizing beam-splitter.

The setup includes three FPCs. The first one is placed before the input coupler (FPC$_{in}$) to align the polarization state of the driving field with one of the two principal polarization modes of the resonator, henceforth defining the $x$ mode. Note that because of residual birefringence, these principal modes correspond in practice to polarization states, which evolve along the fiber but map onto themselves at each roundtrip. The second FPC, placed inside the resonator (FPC$_\pi$), defines the $\pi$ phase-shift defect. We adjust it for $\delta_\pi$ to be as close as possible to zero, which is controlled by monitoring the linear cavity resonances while scanning the driving laser frequency. Finally, the third FPC, placed on the output port of the resonator (FPC$_{out}$) and followed by a polarization beam splitter (PBS), can be set so as to split the output field either in terms of the $x$ and $y$ modes or in terms of the + and − hybrid modes. The intensity of the two components is then recorded with individual 12.5 GHz photodiodes. The total intensity is recorded separately.

To proceed, we record the output intensity levels across our driving pulses over subsequent resonator roundtrips using a real-time oscilloscope. A typical measurement is shown as color plots in Fig. 3a–c in the form of a vertical concatenation (bottom-to-top) of a 20-round trip sequence of real-time oscilloscope traces plotted against time. Panels a and b correspond to the intensities of the + and − hybrid modes, while (c) is the total intensity. The measurements have been obtained with 11 W of peak driving power ($X = 50$) and a roundtrip phase detuning locked to $\delta_0 = 1.16$ rad ($\Delta = 10$). Figure 3a–b reveals that under these conditions, the hybrid mode intensity levels exhibit a broken symmetry ($|\psi_+|^2 \neq |\psi_-|^2$) as well as anti-phase alternating dynamics: the hybrid modes appear to exchange their intensities from one roundtrip to the next. This alternation is a key signature of the modal Möbius topology. At the same time, the total intensity [Fig. 3c] appears constant and presents no sign of the periodic alternation of the hybrid modes. This points to a high level of symmetry between the two hybrid modes and confirms the realization of the $\psi_+ \rightleftharpoons \psi_-$ exchange symmetry. We must note that this remarkable level of symmetry is achieved without any fine-tuning of parameters and can persist for hours. Numerical simulation results presented as Supplementary Section I and Supplementary Fig. S1 are found to be in excellent agreement with all these observations.

Further insights can be gained by measuring the high and low-intensity levels of the hybrid modes for a range of resonator detunings. The results of such measurements are plotted in Fig. 3d for $\Delta$ ranging from −4 to 10 and for the same driving power level as Fig. 3a–c. Blue

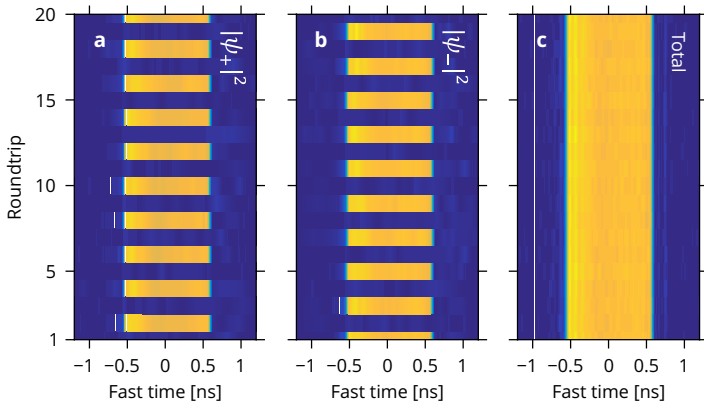

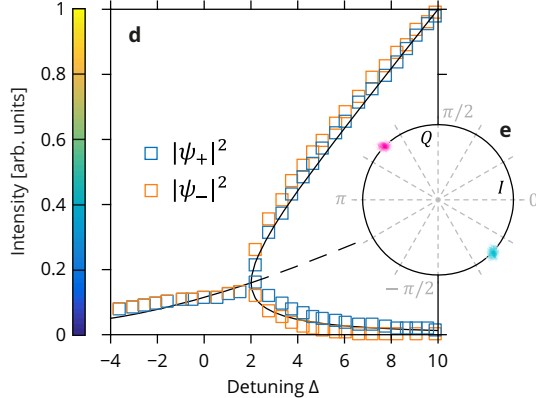

**Fig. 3 | Evidence of P2 SSB.** Oscilloscope recordings of the intensity of the **a** + and **b** − hybrid modes and **c** total intensity over 20 subsequent resonator roundtrips ($X = 50$, $\Delta = 10$). The alternation of the hybrid mode intensities is a key signature of the underlying Möbius topology while the constant total intensity indicates a high level of symmetry between the hybrid modes. **d** High and low-intensity levels of the + (blue) and − (orange) hybrid modes versus resonator detuning, revealing a pitchfork bifurcation ($X = 50$). The black lines are numerical predictions (dashed lines correspond to unstable states). **e** Complex plane showing the real (in-phase, $I$) and imaginary (quadrature, $Q$) components of $\psi_x \psi_y^*$ measured over 1000 resonator roundtrips in the P2 SSB regime. Cyan and purple density plots correspond, respectively, to odd and even roundtrips, and their diametrically opposite placement signals that the relative phase between the $x$ and $y$ fields changes by $\pi$ at every roundtrip. The spread of each density is magnified four times for clarity.

and orange markers correspond to the + and − hybrid modes, respectively. For low values of the detuning, the two hybrid modes are found to have the same intensity, and the intracavity field appears symmetric in terms of these modes. In contrast, beyond $\Delta \simeq 2$, we observe instead two mirror-like, asymmetric states, whose relative contrast increases with the detuning. The emergence of these asymmetric states is found to correspond to the start of the roundtrip-to-roundtrip alternating dynamics described above. These measurements agree with numerically calculated steady-state solutions of Eqs. (13) and (14), which are shown as black curves in Fig. 3d (dashed lines correspond to unstable solutions; discrepancies can be attributed to uncertainties in the experimental parameters). Overall, the data in Fig. 3d exhibit the behavior of a pitchfork bifurcation characteristics of SSB. This result is remarkable in that it is obtained without having to make any effort to realize an exchange symmetry. In fact, we observe this behavior as soon as the $y$-mode phase defect is sufficiently close to $\pi$ (see Supplementary Section II and Supplementary Fig. S2 for more details on the existence range of P2 SSB).

To complete these observations, we have also made measurements in terms of the $x$ and $y$ modes. The emergence of the P2 SSB regime is characterized by the $y$-mode intensity rising above zero through parametric generation (see also Supplementary Section III and Supplementary Fig. S3), but the intensities of the $x$ and $y$ modes do not otherwise reveal the alternating dynamics characteristics of the underlying Möbius topology. As described above, the associated dynamics for the $y$-mode should be entirely contained in a roundtrip-to-roundtrip phase step of $\pi$, which calls for a phase-sensitive measurement. To that end, we have used homodyne detection by mixing the $x$ and $y$ output fields into a single-polarization 90° optical hybrid (Kylia). A pair of 40 GHz-bandwidth balanced detectors then provide in-phase, $I \propto \mathrm{Re}(\psi_x \psi_y^*)$, and quadrature, $Q \propto \mathrm{Im}(\psi_x \psi_y^*)$, components. Measurements of $I$ and $Q$ over 1000 resonator roundtrips in the P2 SSB regime are reported in Fig. 3e, with cyan (respectively, purple) points corresponding to odd (even) roundtrips. The arrangement of these two groups of points in diametrically opposite positions in the complex plane reveals unequivocally that the relative phase between the $x$ and $y$ fields varies in the step of $\pi$ at each roundtrip. The average phase step is found to be $(0.995 \pm 0.020)\pi$. Again, this has been obtained without fine tuning of parameters and confirms that the system is nonlinearly attracted to a roundtrip phase jump of $\pi$ as predicted by theory. Note that, in comparison with the rest of the observations reported in Fig. 3, the data in Fig. 3e was obtained with a slightly shorter but otherwise identical resonator (resonator length of 10.5 m, roundtrip time $t_R = 50.6$ ns, and resonance linewidth $\Delta\omega/(2\pi) = 825$ kHz).

## Test of the exchange symmetry

The results presented in the previous Section provide compelling evidence that a $\pi$-phase defect combined with the Kerr nonlinearity does confer a robust Möbius topology and exchange symmetry to a driven resonator, enabling SSB. In particular, the reproducibility of this behavior and our ability to observe it over long periods of time without any fine-tuning of parameters strongly hint that the exchange symmetry is exact and protected by the nonlinearity, as predicted by theory. In order to confirm this hypothesis further, we now present a more stringent and quantitative test of the quality of the exchange symmetry realized in our system.

Our test is based on the properties of SSB. Specifically, upon crossing the SSB bifurcation threshold in our resonator, an initially symmetric state will lose its stability in favor of one of two states with broken symmetry. Under perfectly symmetrical conditions, this selection should be truly random. Conversely, a deviation from symmetry will lead to a bias in the state selection process[33,35,64]. The degree of randomness of the state selection process is, therefore, a measure of the quality of the exchange symmetry. To test for such randomness in

practice, we sinusoidally modulated the resonator detuning at 5.2 kHz around a mean (locked) value so as to cross the SSB threshold repeatedly and measured the state selection outcome. This measurement is based on the same resonator as that used to obtain the results of Fig. 3e but with a driving beam modulated so as to have 19 pulses simultaneously present in the resonator. Each pulse undergoes SSB independently, providing multiple simultaneous realizations of the experiment, thus enabling the acquisition of a larger amount of data for better statistical significance. The outcomes of the state selection process are determined by analyzing the output pulses captured with our real-time oscilloscope.

For this experiment, we have accumulated a sequence of 2.4 million individual events and expressed them as zero and one binary values. Due to the limitations of our oscilloscope, this data was assembled from four different batches acquired in quick succession over a few seconds without adjusting any parameters. The randomness of the acquired sequence was then rigorously assessed using the NIST Statistical Test Suite for random number generation (NIST STS-2.1.2), which consists of a series of statistical tests used to detect non-randomness within a given data set[65]. In each test, a $p$-value is generated, which gives the probability of this result occurring under the null hypothesis. We consider a $p$-value of 0.01 or lower to be evidence of non-randomness or failure. NIST recommends that the minimum pass rate for each statistical test should be 96% (the whole sequence is partitioned into 100 individual sub-sequences for testing). The results are presented in Fig. 4, which shows that all tests are passed with a proportion well over 96%, highlighting that we have no evidence of any bias in the SSB state selection process. Additional data gathered as the phase defect is varied around $\pi$ further support this finding (see Supplementary Section IV and Supplementary Fig. S4). This confirms the robustness of the exchange symmetry provided by the Möbius topology in our experiment.

## Topological localized structures

In the experimental results discussed so far, we have only considered cases where the intracavity field is homogeneous across the nanosecond driving pulses. The spatial extension of the driving provides, however, the opportunity to support more complex, inhomogeneous field configurations. The P2 SSB regime naturally extends to such structures because the mechanism giving rise to the Möbius topology of our resonator and to the two-roundtrip alternating dynamics is purely local. We now discuss this aspect in more detail.

Inhomogeneous field structures can be accounted for by introducing a kinetic term into the Hamiltonian. This corresponds to adding terms of the form $-i(\phi_2/2)\partial^2\psi_\pm/\partial\tau^2$ to, respectively, the mean-field

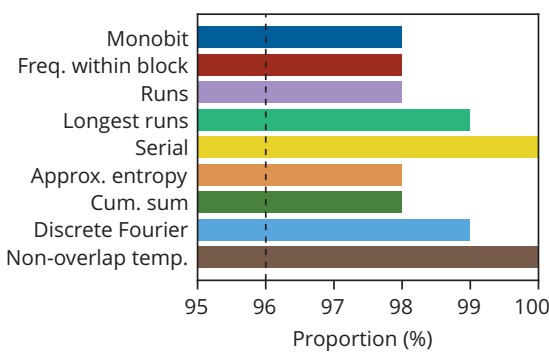

**Fig. 4 | Test results for randomness of the P2 SSB state selection process using the NIST statistical test suite.** The bars indicate the percentage of sequences (a total of 2.4 million events partitioned into 100 sub-sequences) that pass the tests with a significance level of 0.01. The recommended minimum pass rate of 96% (black dotted line) is satisfied for all tests. The normalized driving power was set to $X \simeq 12$.

equation for the + [Eq. (13)] and − [Eq. (14)] hybrid modes. Here $\tau$ is a fast time variable describing variations of the fields along the resonator. In the optical context relevant to our experiments, $\phi_2$ represents chromatic dispersion integrated along the roundtrip of the resonator. With these kinetic terms, Eqs. (13) and (14) take the form of coupled driven-dissipative Gross-Pitaevskii equations. Such equations have been used, e.g., to describe exciton–polariton condensates[38,66,67]. In optics, these equations are known as Lugiato–Lefever equations (LLEs). The LLE was initially introduced as a paradigmatic model of pattern formation in dissipative optical systems[68]. It is also known to support localized structures referred to as cavity solitons[69,70]. In the temporal domain[71], these solitons underlie the new key technology of microresonator broadband optical frequency combs[72,73]. In their coupled form, Eqs. (13) and (14) exhibit an even richer dynamics[62,67,74]. Of particular relevance here are the recent experimental observations of two-component localized structures emerging through SSB, namely domain walls between homogeneous symmetry-broken states[31] and bright symmetry-broken vector cavity solitons[75,76]. Given the formal equivalence of the equations describing these structures and our system, it should be clear that these structures have a P2 counterpart; in fact, the only difference with what was previously reported should be the roundtrip-to-roundtrip alternation of the two components.

We start by considering P2 domain walls, which exist in the presence of repulsive interactions ($\phi_2 > 0$). Experimental evidence is presented in Fig. 5a–c. Panel a shows intensity profiles of the + (blue) and − (orange) hybrid modes at a selected odd (left) and even (right) roundtrip, while panels b and c present the roundtrip-to-roundtrip evolution of those intensities over 20 roundtrips as pseudo-color plots similar to those used in Fig. 3. These plots reveal a field structure whose components are perfectly anti-correlated and, at the same time, alternate roundtrip-to-roundtrip (see also[48]). The driving pulses are subdivided into domains realizing, at any one time, one or the other of the two homogenous symmetry-broken states. By symmetry, the other component has the opposite domain structure. The domains are segregated by sharp kink-like temporal transitions, which correspond to

dissipative (polarization) domain walls or PDWs[31,77,78]. A close-up of a domain wall obtained with a 70 GHz-bandwidth photodiode and sampling oscilloscope is shown in the inset (this measurement is bandwidth-limited with simulations predicting a rise/fall time of 4 ps). These observations were performed with the same resonator and in the same conditions ($X = 50$, $\Delta = 10$) as that used for the results of Fig. 3a–c (for which we have $\phi_2 \simeq + 0.5$ ps/THz). We have only added a shallow 10 GHz sinusoidal phase modulation imprinted onto the driving so as to align and trap the PDWs onto a controlled temporal reference grid[79]. The domain walls were excited spontaneously by ramping up the detuning through the SSB bifurcation threshold. As different parts of the nanosecond driving pulses break their symmetry in different directions, this leads to the emergence of a random pattern of domain walls, which are then trapped by the phase modulation.

Next, we consider P2 bright symmetry-broken temporal cavity solitons. Because these require attractive interactions, i.e., a negative effective mass or, equivalently, anomalous chromatic dispersion ($\phi_2 < 0$), we have used here a different resonator than for all the other observations reported so far in this Article. The resonator was 86 m long and entirely made up of standard SMF28 (Corning) optical fiber with an input coupler that recirculates 95% of the light ($\theta = 0.05$). The roundtrip time $t_R = 420$ ns, the resonance linewidth $\Delta\omega/(2\pi) = 57$ kHz, and $\phi_2 = − 1.7$ ps/THz. The driving and detection schemes were the same as before. Experimental results are presented in Fig. 5d–f using the same layout as for the P2 domain walls. Here, we can observe a sequence of five bright solitons, which are aligned on a 2.87 GHz driving phase modulation grid[79]. Spectral measurements (see Supplementary Section V and Supplementary Fig. S5) indicate a pulse duration of 1.6 ps (not resolved by the oscilloscope). These solitons are symmetry-broken, with one hybrid mode component dominating over the other. Due to the high driving level ($X = 30$, corresponding to 2.5 W peak driving power, and with $\Delta = 14.5$), the contrast between the components is almost 100%. Again, the symmetry implies the existence of two mirror-like states, i.e., solitons exhibiting a different dominant component, and our data confirms that these

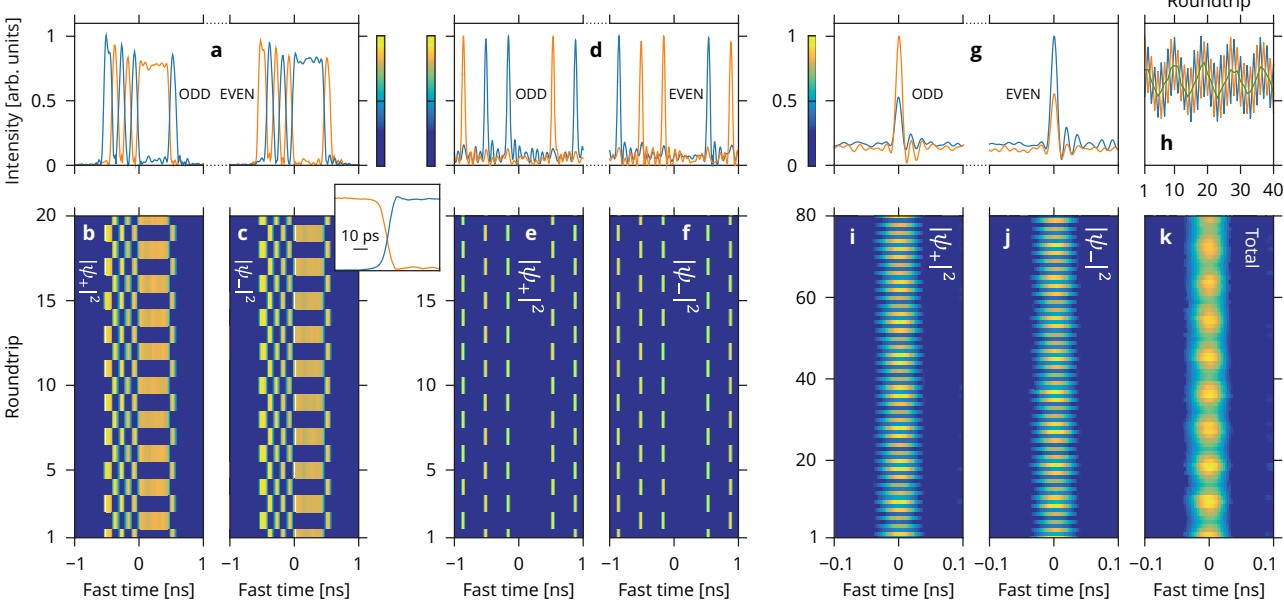

**Fig. 5 | P2 SSB localized structures. a–c** P2 domain walls observed for repulsive interactions ($\phi_2 > 0$) with $X = 50$ and $\Delta = 10$. **a** Temporal intensity profiles of the + (blue) and − (orange) hybrid modes at a selected odd (left) and even (right) roundtrip. The evolution of these profiles over 20 subsequent resonator roundtrips is illustrated in (**b**, **c**), respectively, for the + and − hybrid modes. Inset: close up on one of the domain walls. **d–f** same as **a–c** but for P2 bright symmetry-broken cavity

solitons observed for attractive interactions ($\phi_2 < 0$) with $X = 30$ and $\Delta = 14.5$. **g–k** Breathing dynamics of the P2 bright symmetry-broken cavity solitons observed with $X = 10$ and $\Delta = 5.2$. Same format as the other two examples with the addition of (**k**) an extra pseudo-color plot for the evolution of the total intensity and (**h**) a plot providing details of the evolution of the intensity at the center of the structure (blue/orange, +/− hybrid mode; green, total intensity, halved).

simultaneously coexist in the resonator. Finally, we observe a regular alternation of the components, with + mode-dominant solitons becoming − dominant and vice versa, each roundtrip to the next, as expected from the Möbius topology.

Bright symmetry-broken cavity solitons are also known to go through a Hopf bifurcation for lower detuning values, leading to breathing dynamics. This dynamic is characterized by slow in-phase oscillations of the intensities of the two components of the soliton[76]. In Fig. 5g–k, we provide evidence of this dynamic in the P2 SSB regime. As for the other structures discussed above, panels i and j show color plots of the evolution of the + and − hybrid mode temporal intensity profiles. Additionally, panel k displays the corresponding evolution for the total intensity. The latter clearly reveals the breathing dynamics occurring here with a period of about 9 resonator roundtrips, while the roundtrip-to-roundtrip alternation characteristics of the P2 SSB regime are only detected in the individual components. The superposition of breathing and alternation is better seen in panel h, with curves highlighting the evolution at the center of the soliton in terms of + (blue), − (orange), and total (green) intensities; these correspond to cross-sections through the center of panels i–k, respectively. Note that we have selected here a lower driving level ($X = 10$) than in Fig. 5d–f, with the detuning set to $\Delta = 5.2$. This choice leads to less contrast between the soliton components [compare Fig. 5d and g], and provides a clearer illustration of the overall complexity of the P2 symmetry-broken breather dynamics.

Finally, we must point out that all the structures reported in Fig. 5 could consistently be maintained and observed for long periods of time, from 30 to 60 min (see Supplementary Section VI and Supplementary Figs. S6 and S7). This confirms that the robustness of the Möbius topology extends to complex inhomogeneous structures.

## Discussion

We have described theoretically and experimentally a system characterized by a complex interplay between topology, nonlinearity, and SSB. Specifically, we have shown that a $\pi$ phase defect can force a driven Kerr resonator with two modes, $x$ and $y$, toward an attractor arising through nonlinear degenerate parametric interactions so that the system displays a modal Möbius topology. That topology is associated with an exchange symmetry, which, together with the nonlinearity, enables SSB. Because of the Möbius topology, the two symmetry-broken states of our resonator alternate roundtrip-to-roundtrip, which we refer to as P2 SSB. In the experiment, this alternation is directly seen in the intensity of the + and − hybrid modes. Homodyne measurements have also enabled us to detect the corresponding phase steps in the evolution of the $x$ and $y$ modes of the resonator. Because the Möbius topology arises through a purely local mechanism, it extends to inhomogeneous field distributions, giving rise to a range of P2 localized structures, which we have also been able to observe. Thanks to the robustness associated with the topology, SSB is realized without any bias or fine-tuning of parameters. This fact has been theoretically confirmed by an effective Floquet Hamiltonian model and experimentally tested through rigorous testing of the randomness of the symmetry-broken state selection statistics. The inherent robustness is also seen in the fact that all the different behaviors we have reported can be obtained without any sensitive adjustments of parameters and are consistently observed over long periods of time, easily exceeding 30 min (typically only limited by our detuning locking feedback loop). As the realization of SSB is often hampered by biases and non-idealities, we believe that our findings can have interest for a number of applications, in photonics, but more broadly in other bosonic systems, including two-component atomic gases, cold bosonic atoms trapped in a double well, or spinors (see, e.g.,[41,80]).

As a final remark, we would like to mention that phase-defect-induced topological protection can be generalized to systems with more than two modes, potentially leading to P3, P4, … regimes of SSB.

In what follows, we present brief arguments to illustrate the general idea, restricting our attention to three modes for simplicity. Consider a resonator with modes $x$, $y$, and $z$, where the $y$ and $z$ modes present phase defects of about, respectively, $2\pi/3$ and $4\pi/3$ with respect to the $x$ mode. This could be implemented, e.g., using etched waveguide-based adiabatic mode converters or photonic crystals[50,81]. The resonance frequencies of the three mode families would be shifted in steps of a third of the FSR. With these modes, it is possible to define hybrid modes $\alpha$, $\beta$, $\gamma$, such that the hybrid mode amplitudes would be converted at each roundtrip $\alpha \to \beta \to \gamma \to \alpha$, and display a generalized Möbius topology over a cycle of three resonator roundtrips. This will realize a cyclic $\mathbb{Z}_3$ symmetry. The nonlinear four-wave interactions enabled by energy conservation are of the form $x + x \to y + z$ (and permutations). Using the same approach as in the theory presented above, it is easy to show that these interactions make the $2\pi/3$ and $4\pi/3$ roundtrip phase steps robust attractors of the system, making the $\mathbb{Z}_3$ symmetry exact, and protecting the topology. We have further confirmed that parameters exist where this system exhibits three symmetry-broken states, leading to P3 SSB, where these states cycle over three roundtrips.

As a particular application, we must note that the symmetry-broken states described in our Article could potentially be used to realize artificial spins. In the P2 SSB regime, these would correspond to spin-1/2 and could be exploited to implement an Ising machine. Such a machine aims to find the ground state of interacting spins described by the Ising Hamiltonian with an appropriate physical system. The interest rests on the fact that many complex combinatorial optimization problems relevant to modern society, such as drug discovery[82] or the analysis of social networks[83], can be mapped onto an Ising Hamiltonian[84,85]. We must note that coherent optical Ising machines based on degenerate parametric oscillation have been particularly successful in recent years[36,37]. A P2 SSB Ising machine would work along the same principle, but the inherent topological robustness of the Möbius topology could provide additional benefits. In this context, using more than two modes would offer even more tantalizing possibilities. For example, P3 symmetry-broken states would correspond to spin-1 particles and open the way to the realization of a Potts machine[86].

## Methods
### Classical derivation
We present here a classical derivation of the mean-field equations (13) and (14). This derivation complements our Hamiltonian description, which was derived in the conservative limit, by including dissipation and driving explicitly. For completeness, we also consider modes with arbitrary polarization states, i.e., arbitrary values of the $A$, $B$, and $C$ coefficients [see Eq. (2)]. We start from the classical equations of motion for the two modes, i.e., Eq. (3) and a similar equation for $\psi_x$. These equations are complemented by boundary conditions,

$$\begin{pmatrix} \psi_x \\ \psi_y \end{pmatrix}^{(m+1)} = e^{-\alpha} M \begin{pmatrix} \psi_x \\ \psi_y \end{pmatrix}^{(m)} + \sqrt{\theta}\, S_{\text{in}} \begin{pmatrix} \cos\chi \\ \sin\chi \end{pmatrix}. \quad (15)$$

Here, $\alpha = t_R \Delta\omega/2$ is the dissipation rate of the resonator, as introduced before, $S_{\text{in}}$ is the total driving amplitude, with the ellipticity angle $\chi$ describing a split of the driving between the two modes ($\chi = 0$ corresponds to only driving the $x$ mode as previously considered, and $S_x = S_{\text{in}} \cos\chi$), and the matrix $M$ represents the action of the detunings,

$$M = \begin{pmatrix} e^{-i\delta_0} & 0 \\ 0 & e^{i\pi} e^{-i(\delta_0 - \delta_\pi)} \end{pmatrix}. \quad (16)$$

We proceed by restricting ourselves to high-Q resonators operated close to resonance and by following an approach similar to that of

Refs. [87],[88]. In these conditions, $\alpha$, $\theta$, $\delta$, and $\delta_\pi$ can be assumed $\ll 1$. Similarly, all the nonlinear terms in the equations of motion are small, and these equations can be integrated over one roundtrip time $t_R$ by a single Euler step, as we have done with Eq. (4). Combining the result with the boundary conditions, Eq. (15), and expanding all small terms at first order, we can express the fields at roundtrip $m+1$ in terms of those at roundtrip $m$ as

$$\psi_x^{(m+1)} = \left[1 - \alpha - i\delta_0 + igt_R\left(A|\psi_x^{(m)}|^2 + B|\psi_y^{(m)}|^2\right)\right]\psi_x^{(m)}$$
$$+ igt_R C\,\psi_x^{(m)*}\psi_y^{(m)2} + \sqrt{\theta}\,S_{in}\cos\chi, \tag{17}$$

$$\psi_y^{(m+1)} = -\left[1 - \alpha - i(\delta_0 - \delta_\pi) + igt_R\left(A|\psi_y^{(m)}|^2 + B|\psi_x^{(m)}|^2\right)\right]\psi_y^{(m)}$$
$$- igt_R C\,\psi_y^{(m)*}\psi_x^{(m)2} + \sqrt{\theta}\,S_{in}\sin\chi. \tag{18}$$

Above, we can observe that the leading contribution for the $y$ mode is a change of sign, as expected from the $\pi$ phase defect.

The next step is to iterate Eqs. (17) and (18) and seek expressions for $\psi_{x,y}^{(m+2)}$ in terms of the $m$th roundtrip fields. At first order, we can assume that, in the nonlinear terms that result, $\psi_x^{(m+1)} \approx \psi_x^{(m)}$ and $\psi_y^{(m+1)} \approx -\psi_y^{(m)}$. Because of the double action of the $\pi$ phase defect over two roundtrips, we then find that the changes in the fields over two roundtrips, $\psi_{x,y}^{(m+2)} - \psi_{x,y}^{(m)}$, are first order quantities. This enables us to introduce a slow-time derivative over two resonator roundtrips, defined as $d./dt = \left[.^{(m+2)} - .^{(m)}\right]/(2t_R)$, which transforms the difference equations we get into the following differential equations (this latter step is analogous to the stroboscopic Floquet approach used in the main Article),

$$t_R\frac{d\psi_x}{dt} = \left[-\alpha - i\delta_0 + igt_R\left(A|\psi_x|^2 + B|\psi_y|^2\right)\right]\psi_x$$
$$+ igt_R C\,\psi_x^*\psi_y^2 + \sqrt{\theta}\,S_{in}\cos\chi, \tag{19}$$

$$t_R\frac{d\psi_y}{dt} = \left[-\alpha - i(\delta_0 - \delta_\pi) + igt_R\left(A|\psi_y|^2 + B|\psi_x|^2\right)\right]\psi_y$$
$$+ igt_R C\,\psi_y^*\psi_x^2. \tag{20}$$

where we have dropped the roundtrip index $m$. Here we can observe in particular that the $y$ component of the driving cancels out over two roundtrips. This stems from that contribution being close to anti-resonance conditions due to the half-FSR shift between the $x$ and $y$ mode resonances. Accordingly, the driving ellipticity $\chi$ simply affects the effective driving power, and any misalignment of the driving polarization with that of the $x$ mode can simply be compensated for by driving the resonator stronger.

The last step is to express the equations above in terms of the $+$ and $-$ hybrid mode amplitudes defined by Eqs. (1). This leads to Eqs. (13) and (14), with

$$A' = \frac{1+B-C}{2}A, \qquad B' = (1+C)A, \tag{21}$$

but with additional parametric terms, $igt_R C'\psi_+^*\psi_-^2$ for Eq. (13) and $igt_R C'\psi_-^*\psi_+^2$ for Eq. (14), where

$$C' = \frac{1-B-C}{2}A. \tag{22}$$

We can observe that these terms do not break the $\psi_+ \rightleftarrows \psi_-$ exchange symmetry of the equations and overall do not change the phenomenology. When the $x$ and $y$ modes correspond to linear polarization states, $C' = 0$ as in Eq. (6).

Note that the above analysis assumes a $\pi$ phase defect that is purely localized. Additional considerations show that distributing the phase defect over a certain fraction of the resonator length amounts to a phase mismatch for the parametric interaction (nonconservation of momentum). This leads to less efficient generation of the $y$ mode, but P2 SSB remains possible. In our experiments, $FPC_\pi$ typically uses about one meter of fiber, so it is distributed over about 10% of the resonator or less (depending on the resonator length). Simulations indicate that P2 SSB can persist even with the phase defect distributed over more than half the resonator length but typically requires an increase in driving power. When parametric generation cannot overcome the losses of the resonator anymore, the P2 SSB regime eventually disappears.

Kinetic terms (chromatic dispersion) can also be included in our derivation in a straightforward manner. We note that neither these terms nor any higher-order terms like third-order dispersion or stimulated Raman scattering that lead to asymmetries in the fast-time variable $\tau$, affect the exchange symmetry of the hybrid modes.

## Data availability
The data that support the plots within this paper and other findings of this study are available from the corresponding author upon request.

## Code availability
The code that supports the plots within this paper and other findings of this study are available from the corresponding author upon request.

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

## Acknowledgements

We acknowledge financial support from The Royal Society Te Apārangi in the form of Marsden Funding (18-UOA-310) as well as M.E.'s Rutherford Discovery Fellowship (RDF-15-UOA-015). J.F. thanks the Conseil régional de Bourgogne Franche-Comté, mobility (2019-7-10614), financial support from the CNRS, the IRP Wall-IN project, and FEDER. N.G. is supported by the FRS-FNRS (Belgium), the EOS program (CHEQS project), and the ERC (TopoCold and LATIS projects).

## Author contributions

B.G. and J.F. set up the experiment and made the initial observations of P2 SSB and P2 domain walls. J.F. carried on subsequent numerical modeling and made the measurements shown in Fig. 3, as well as the P2 domain walls [Fig. 5a–c]. G.X. performed the experiment related to the P2 bright symmetry-broken cavity solitons and associated breather [Fig. 5d–k]. L.Q. was experimentally implemented and carried out on the randomness statistical test of Fig. 4. N.G. derived the Hamiltonian model. G.L.O., M.E., and S.M. provided extra theoretical and/or experimental support. S.C. suggested the topological interpretation of P2 SSB and derived the classical mean-field description. All authors helped in interpreting the results. S.C. wrote the paper with feedback from all the authors.

## Competing interests

The authors declare no competing interests.
