## [Peer Review File · Nature Communications]

REVIEWER COMMENTS

Reviewer #1 (Remarks to the Author):

This paper illustrates an experimental observation of symmetry-breaking characteristics in a system with a synthetic Möbius topology. The main observation made by the authors is that due to the nonlinearity of their experimental resonator-based system, which leads to the formation of attractors, the system is remarkably robust and does not require extremely fine parameters tuning to obtain symmetry-breaking observations. Furthermore, thanks to a derivation of the Hamiltonian of this 2D system, they provide an in-depth theoretical support to explain the system's symmetry breaking and its robustness due to attractors. They provide a clear observation of the symmetry-breaking characteristics by adding a localized π shift in a fiber ring resonator. The results are very clear-cut, achieved over a wide range of parameters to reveal this observation in both attractive and repulsive regimes. The "quality" of the randomness between the two states is validated using the NIST statistical test suite, a standard test in the field. Finally, an interesting discussion of the potential of these systems for >2 dimensions is provided to realize artificial spins.

The article is very well written and illustrated. The interest of the work, concerning the sensitivity to imperfections in other systems that affect symmetry breaking, is well presented and understandable even to non-specialists. The experimental results are very convincing and clearly presented. There is no doubt that this is an excellent work that deserves to be published. I just have one small concern about the journal, I think it would be more relevant in Nature Physics rather than Nature communications. There's a lot of physics in this paper, original concepts of mixing attractors, Möbius characteristics and topology, which seems more suited to Nat. Phys. In any case, I recommend publication of this paper, which would make an excellent article for Nat. Comm.

I have the following questions before publishing the article:

- I expect excellent agreement with numerics, would it be possible to add numerical simulations in the supplemental part of this paper. This would confirm even more the observations and strengthen the results.
- The system's attractors make it almost insensitive to imperfections, which is clearly illustrated by the experiments. Would it be possible, either numerically or experimentally, to illustrate this by the evolution of one of the NIST test parameters (or any other parameter illustrating SSB quality) as a function of one of the system parameters (π offset, ...)?
- The dynamics of the system is expected to be described by the set of coupled equations 13-14. There are no higher order terms, such as Raman or β_3 which are known to affect the symmetry of these systems. Is it possible to comment on that?
- In the theoretical analysis, I would put all the details of the derivations in Methods, which would lighten the reading of this part, which is only interesting for experts.

- The authors have assumed high-Q resonators to simplify equation (4). What is the limit of this assumption? Does it affect SSB?

- I'm not a theorist. I know the Floquet analysis, but I don't know the Floquet Hamiltonian. Is this the Floquet analysis of this Hamiltonian system?

- Some of these observations are very similar to domain wall observations on these systems (see references in the article). Could we simply say that this is another interpretation of this process?

Reviewer #2 (Remarks to the Author):

The manuscript written by S. Coen et al. demonstrates a robust and bias-free spontaneous symmetry breaking by introducing a novel Mobius topology in a two-mode system. The Mobius topology – realized by π -phase-shift defect between two linear polarization modes – enables swapping of the hybrid mode every roundtrip, facilitating exchange symmetry for the hybrid modes. The work establishes a theoretical framework for realizing robust symmetry and experimentally validates the concept, underlined by a statistical randomness test. The demonstrated experiments are not significant in novelty, but they rigorously validate the claimed robustness in anomalous and normal dispersion regimes with various temporal structures. The significance of the work lies in the realization of symmetry protection which fundamentally sets the work apart from the previous works on spontaneous symmetry breaking in photonics.

On this basis, I find the manuscript suitable for publication in Nature Communications after answering the following questions.

1) For the sake of the reproducibility of the work and extending its reach to other platforms (i.e., integrated photonics), adding more discussion about the x/y mode and nonlinear phenomena would be beneficial. I understand that the x/y-mode is less relevant to the flow of the manuscript and can be assumed based on previous works on SSB. However, this work differs from the traditional SSB experiments which may trigger curiosity. Please consider this a set of questions that the author may freely decide whether or not to reflect in the manuscript upon answering.

a. Since the work introduces an intentionally detuned y mode resonance, I would like to know where the generated y mode is spectrally located (closer to the x mode resonance or the FSR/2 shifted y mode resonance).

b. The experiment result does not show x/y-mode evolution over the detuning, however, there is a conceptual plot in Fig. 1(c). Especially, here, the y-mode is the critical component in realizing SSB. In reality, does the y-mode intensity show monotonous evolution over the detuning like Fig. 1(c)? In connection to this, what is the reason behind not including a plot of x/y-mode evolution like Fig. 1(c)?

c. Does the nonlinear resonance shift (at different pump power levels) of the x mode resonance influence the outcome? The driven x-mode resonance will initially have a larger resonance shift than the π -shifted y-mode resonance. This question can be disregarded if the y-mode resonance does not hold significance in the operation.

d. Is there a sweet spot in terms of the operating parameters (e.g., driving power)? In connection with that, are there any other nonlinear phenomena that had to be suppressed or avoided in realizing the experiment? What were the measures taken to address these?

e. Achieving both (a) linear coupling free polarization condition and (b) exact π phase shift for the y mode – at the same time – do not seem trivial (FPC_ π). Could you elaborate if there was any efficient FPC manipulation method that is not mentioned in the manuscript?

2) π phase defect and the experiment regarding the exchange symmetry are the key elements of the work. I would like to read more about the details.

a. Was there any failing condition for the statistical randomness test (exact symmetry) in the experiment or numerical simulation? How tolerant is the experiment in regards to different y-mode phase shifts (0 to π) when driving in the x-mode? Also, does the initial ellipticity of the pump influence the result? These questions can be answered quantitatively or qualitatively.

b. “Over multiple runs, we have accumulated a sequence of 2.4 million individual events”

Why did it require multiple runs to record the events instead of a single continuous measurement? How long in total does it take to record 2.4 million events? Did the authors have to readjust some parts of the system in between runs?

c. “In fact, we observe this behavior as soon as the y-mode phase defect is sufficiently close to π (to within about the resonance linewidth).”

From the sentence above, can the condition “within about the resonance linewidth” be generalized for other systems? If not, could you elaborate on what needs to be considered?

d. “Note that the above analysis assumes a π phase defect that is purely localized.” Considering the sentence above, mentioning the phase defect-to-cavity length ratio in the experiment would be helpful. Also, at what ratio will the y-mode intensity halve? Please answer if this is something trivial to extract. Although these are specific to the demonstrated experiment, this may assist audiences’ initial attempt to choose cavity length and phase defect component (mode converter or FPC).

Minor comments:

1) Fig. 3 y-label: Intensity [a.u.] -> Intensity [a.u.]

2) Regarding SSB, I noticed a recent work [Moroney, Niall, et al. "A Kerr polarization controller." Nature Communications 13.1 (2022): 398.]. Exposing the audience to a different take (exploiting the asymmetry for deterministic polarization control) on the polarization mode SSB would be beneficial. *Only if the authors agree that it is relevant to include it in the manuscript.

Reviewer #3 (Remarks to the Author):

The paper presents a novel exploration into the fascinating interplay between topology, nonlinearity, and spontaneous symmetry-breaking (SSB) in the realm of driven Kerr resonators. The authors' attempt to demonstrate this complex relationship is commendable and highlights a significant step forward in the research in this field.

1. The authors have constructed a robust theoretical framework, pioneering the concept of a Möbius topology and linking it with SSB phenomena. However, some questions remain: The authors should comment on the effectiveness of their model at negative detuning, as the fitting to the experimental data becomes less accurate.

2. The authors should provide more experimental evidence of the symmetry protection and stability. The authors refer to 30min stability limited only by the locking feedback loop, which I argue is also part of the system. I understand that this is not inherent to the SSB in the ring resonator, but more results, or at least arguments on the thermal and polarization stability in the fiber loop should be given.

3. The presented evidence, compellingly supports the existence of P2 domain walls - a noteworthy finding attributed to the Möbius topology. However, the connection between this unique topological feature and the emergence of these domain walls requires further clarification.

4. The authors indicate the implications of their work might extend beyond photonics into other bosonic systems. While this proposition is intriguing, it would be beneficial to provide explicit examples of practical applications that could potentially be influenced by these findings. Doing so would underline the novelty and significance of their research. The work present great fundamental physics investigation, perhaps more emphasis on the applications is appreciated.

5. An interesting claim made by the authors is the extension of the robustness associated with the Möbius topology to complex inhomogeneous structures. This assertion could benefit from more supportive evidence, be it empirical or theoretical.

In summary, the manuscript provides a considerable and innovative contribution to its field and is well suited for publication in Nature Communications. I recommend acceptance, provided the authors address the queries and suggestions raised in their revised submission.

Response to Reviewer #1

This paper illustrates an experimental observation of symmetry-breaking characteristics in a system with a synthetic Möbius topology. The main observation made by the authors is that due to the nonlinearity of their experimental resonator-based system, which leads to the formation of attractors, the system is remarkably robust and do not require extremely fine parameters tuning to obtain symmetry-breaking observations. Furthermore, thanks to a derivation of the Hamiltonian of this 2D system, they provide an in-depth theoretical support to explain the system's symmetry breaking and its robustness due to attractors. They provide a clear observation of the symmetry-breaking characteristics by adding a localized π shift in a fiber ring resonator. The results are very clear-cut, achieved over a wide range of parameters to reveal this observation in both attractive and repulsive regimes. The "quality" of the randomness between the two states is validated using the NIST statistical test suite, a standard test in the field. Finally, an interesting discussion of the potential of these systems for > 2 dimensions is provided to realize artificial spins.

The article is very well written and illustrated. The interest of the work, concerning the sensitivity to imperfections in other systems that affect symmetry breaking, is well presented and understandable even to non-specialists. The experimental results are very convincing and clearly presented. There is no doubt that this is an excellent work that deserves to be published. I just have one small concern about the journal, I think it would be more relevant in Nature Physics rather than Nature communications. There's a lot of physics in this paper, original concepts of mixing attractors, Möbius characteristics and topology, which seems more suited to Nat. Phys. In any case, I recommend publication of this paper, which would make an excellent article for Nat. Comm.

I have the following questions before publishing the article:

- I expect excellent agreement with numerics, would it be possible to add numerical simulations in the supplemental part of this paper. This would confirm even more the observations and strengthen the results.

We have added simulations of P2 SSB as Supplementary Section I. These have purposefully been done with a deviation from an exact π phase defect ($\delta_\pi \neq 0$) to confirm the robustness of the P2 alternating dynamics in such conditions. These simulations are referred to in Section IV of the main Article when discussing the results presented in Fig. 3.

- The system's attractors make it almost insensitive to imperfections, which is clearly illustrated by the experiments. Would it be possible, either numerically or experimentally, to illustrate this by the evolution of one of the NIST test parameters (or any other parameter illustrating SSB quality) as a function of one of the system parameters (π offset, ...)?

We thank the Reviewer for this suggestion. We have now taken extra experimental measurements of the randomness of the state selection process as a function of the π offset, δ_π . These are reported as Supplementary Section IV (and referred to as at the end of Section V of the main Article) in the form of p -values for the monobit test (whether the two states appear with the same likelihood, as expected for a truly random process) plotted against δ_π . The p -values obtained are nicely clustered in the upper range of values (> 0.01), indicating randomness, except for the largest negative value of δ_π considered. For the latter, the test fails because we are too

close to the threshold of P2 SSB, and cannot reliably distinguish between the two states anymore. See also our reply to comment 2 of Reviewer 2.

- The dynamics of the system is expected to be described by the set of coupled equations 13-14. There are no higher order terms, such as Raman or beta3 which are known to affect the symmetry of these systems. Is it possible to comment on that?

Raman and third-order dispersion lead to asymmetries but only along the fast time direction τ that describes variations of the field along the resonator. However, as explained in our manuscript, the mechanism that gives rise to the Möbius topology and to the alternating dynamics is purely local and plays out over successive roundtrips (slow time) rather than along the fast time dimension. Hence, the alternating dynamics is not affected by Raman and third-order dispersion, and the symmetry between the two hybrid modes E_{\pm} is preserved even in presence of these terms. We have added a comment about this point at the end of the classical derivation in the Methods.

- In the theoretical analysis, I would put all the details of the derivations in Methods, which would lighten the reading of this part, which is only interesting for experts.

We have considered this point carefully and in the end we have decided to keep the presentation as in the original submission. First, we note that the other two Reviewers did not raise any concern regarding this point. Second, we note that the theoretical section contains two key results, namely the demonstration of the stability of the roundtrip-to-roundtrip π phase step and the proof of the symmetry of the system. While these could be summarized, with details put in the Methods, we feel that overall this would weaken the presentation.

- The authors have assumed high-Q resonators to simplify equation (4). What is the limit of this assumption? Does it affect SSB?

This assumption has been made to simplify the presentation but even under more general (lower Q) conditions, where the integration of Eq. (4) would be more complicated, the general mechanism that leads to the robustness of the π phase defect, as described below Eq. (5), would still hold. As long as the nonlinear phase shift accumulated over one roundtrip is sufficiently distinct from π , or in other words, as long as the nonlinearity does not lead to an overlap of adjacent resonances, we expect the π phase defect to be robust. This is for example confirmed by the new numerical simulations presented as Supplementary Section I (first point raised by this Reviewer) which have been performed with a lumped map model of the resonator rather than using the mean-field equations. We have now clarified in our revised manuscript that the assumption of high-Q is made to simplify the discussion.

- I'm not a theorist. I know the Floquet analysis, but I don't know the Floquet Hamiltonian. Is this the Floquet analysis of this Hamiltonian system?

The two are related. The Floquet analysis usually refers more specifically to a form of linear stability analysis of the solutions of time-periodic systems (such as periodically-driven quantum systems). More generally, the Floquet Hamiltonian is concerned with a stroboscopic description of the dynamics of such systems over long time scales (long compared to the driving period). The latter is achieved by the introduction of an effective (time-independent) Hamiltonian operator — often known as “Floquet Hamiltonian.” This effective Hamiltonian is defined by application of the time-evolution operator of the system over one period of the time-periodic modulation [in our case, Eq. (9)].

- Some of these observations are very similar to domain wall observations on these systems (see references in the article). Could we simply say that this is another interpretation of this process?

Domain walls are inhomogeneous structures characterized by a connection between two symmetry-broken states. It is therefore not surprising to find some domain wall-type structures in our current system since it is underlined by a symmetry breaking bifurcation. However, the alternating P2 mechanism is more general than that and is not directly related to domain walls, and indeed we have also reported alternating bright symmetry-broken cavity solitons and breathers which have nothing to do with domain walls. So the answer to the Reviewer's question is no: the alternating P2 dynamics is not another interpretation of domain walls. Rather, domain walls, among a range of other structures, benefit from a much enhanced robustness from the alternating P2 dynamics.

Response to Reviewer #2

The manuscript written by S. Coen et al. demonstrates a robust and bias-free spontaneous symmetry breaking by introducing a novel Möbius topology in a two-mode system. The Möbius topology — realized by π -phase-shift defect between two linear polarization modes — enables swapping of the hybrid mode every roundtrip, facilitating exchange symmetry for the hybrid modes. The work establishes a theoretical framework for realizing robust symmetry and experimentally validates the concept, underlined by a statistical randomness test. The demonstrated experiments are not significant in novelty, but they rigorously validate the claimed robustness in anomalous and normal dispersion regimes with various temporal structures. The significance of the work lies in the realization of symmetry protection which fundamentally sets the work apart from the previous works on spontaneous symmetry breaking in photonics.

On this basis, I find the manuscript suitable for publication in Nature Communications after answering the following questions.

1) For the sake of the reproducibility of the work and extending its reach to other platforms (i.e., integrated photonics), adding more discussion about the x/y mode and nonlinear phenomena would be beneficial. I understand that the x/y-mode is less relevant to the flow of the manuscript and can be assumed based on previous works on SSB. However, this work differs from the traditional SSB experiments which may trigger curiosity. Please consider this a set of questions that the author may freely decide whether or not to reflect in the manuscript upon answering.

a. Since the work introduces an intensionally detuned y mode resonance, I would like to know where the generated y mode is spectrally located (closer to the x mode resonance or the FSR/2 shifted y mode resonance).

The generated y -mode has spectral components shifted by $\pm\text{FSR}/2$ from the driving beam. This is consistent with the Fourier components of a signal that flips its sign ($E_y \Leftrightarrow -E_y$) at every roundtrip. We have added a comment regarding this point near the end of Section II.

b. The experiment result does not show x/y-mode evolution over the detuning, however, there is a conceptual plot in Fig. 1(c). Especially, here, the y-mode is the critical component in realizing SSB. In reality, does the y-mode intensity show monotonous evolution over the detuning like Fig. 1(c)? In connection to this, what is the reason behind not including a plot of x/y-mode evolution like Fig. 1(c)?

The Reviewer is right that the Article mostly focus on reporting measurements for the + and – hybrid modes, because this is where the P2 dynamics, and its relation to symmetry breaking, appears in the most obvious way. However, we appreciate the Reviewer’s concern and we have added as Supplementary Section III measurements of the intensity of the x/y and +/- modes plotted in parallel across a full resonance scan. This constitutes an experimental counterpart of the conceptual plot of Fig. 1(c). These extra measurements are referred to in the last paragraph of Section IV of the main Article.

c. Does the nonlinear resonance shift (at different pump power levels) of the x mode resonance influence the outcome? The driven x -mode resonance will initially have a larger resonance shift than the π -shifted y -mode resonance. This question can be disregarded if the y -mode resonance does not hold significance in the operation.

The nonlinear resonance shift increases with pump power, and since the x and y field components are shifted in frequency by a constant amount (half the free-spectral-range) we can expect a broader range of existence of P2 SSB if the y -mode resonance is offset in the same direction as the nonlinear tilt (corresponding to positive values of δ_π). Indeed, this enables the x and y components to be simultaneously closer to their respective resonances. To illustrate this point, we now report a systematic numerical and analytical study of the P2 SSB existence range as Supplementary Section II (and referred to as in Section IV) which confirms the statement above. Analytical estimates are also found to be in excellent agreement with numerical results. This new data also addresses points 1d and 2a of this Reviewer, below.

d. Is there a sweet spot in terms of the operating parameters (e.g., driving power)? In connection with that, are there any other nonlinear phenomena that had to be suppressed or avoided in realizing the experiment? What were the measures taken to address these?

The range of detunings where SSB occurs widens for higher driving power. At higher power, the contrast between the two symmetry broken states is typically larger [compare, e.g., the difference in contrast between the two hybrid mode intensities in Fig. 5(d) and 5(g)]. Different applications might benefit from different properties but there is no sweet spot. We believe this point is well illustrated in our manuscript, in which we present results obtained with different resonators and at different driving power levels. We also clearly state that no fine tuning of parameter is required. The new numerical and analytical results on the P2 SSB existence range in Supplementary Section II (see points 1c above and 2a below) also addresses this aspect.

No other nonlinear phenomena had to be avoided in realizing the experiment. This point could be related to the question of Reviewer 1 about whether stimulated Raman scattering might introduce an asymmetry, but this is not the case, as the P2 alternating dynamics arises locally, and is not related to any fast time dynamics. This point has been clarified at the end of the Methods.

e. Achieving both (a) linear coupling free polarization condition and (b) exact π phase shift for the y mode – at the same time – do not seem trivial (FPC_π). Could you elaborate if there was any efficient FPC manipulation method that is not mentioned in the manuscript?

There is no complication involved in tuning FPC_π and the points raised by the Reviewer are not critical at all. Linear coupling between the modes is not a problem because the spectral components (and the resonances) of the x and y -modes are distinct (shifted by $FSR/2$; see point 1a above) and therefore cannot linearly couple to each other. We now mention this point below Eqs. (13)–(14). Also, the π phase shift does not have to be exact (see also point 2a below). While monitoring the linear resonances (at low power, with a periodic scan of the detuning), we simply adjust FPC_π so that the y -mode resonances are as close as possible to being half way between the x -mode resonances. This is done by a simple visual estimate on an oscilloscope.

2) the π phase defect and the experiment regarding the exchange symmetry are the key elements of the work. I would like to read more about the details.

a. Was there any failing condition for the statistical randomness test (exact symmetry) in the experiment or numerical simulation? How tolerant is the experiment in regards to different y -mode phase shifts (0 to π) when driving in the x -mode? Also, does the initial ellipticity of the pump influence the result? These questions can be answered quantitatively or qualitatively.

The only case where we fail the randomness test is when experimental conditions are such that the contrast between the two symmetry-broken states is too weak for them to be reliably distinguished. This occurs in particular when the range of detuning over which SSB occurs is small. Also, after a couple of hours of operations, drift in environmental conditions may push the system towards those limits.

With respect to the y -mode phase shift, the system is actually quite tolerant. We have now added as Supplementary Section IV experimental randomness tests of the symmetry-broken state selection process as a function of the π -phase offset, δ_π . This also addresses the second comment of Reviewer 1. The results are compatible with a purely random selection, independently of δ_π , as long as the contrast between the states is good enough.

We have supplemented these measurements by a more systematic numerical and analytical study of the P2 SSB existence range as a function of driving power X and π -phase offset δ_π . This is now presented as Supplementary Section II and also addresses points 1c and 1d above. These new results reveal that the system is actually much more tolerant at higher driving power when considering *positive* values of δ_π . This can be understood by noting that this corresponds to a shift of the y resonance in the same direction as the nonlinear tilt, which enables the x and y components (which are shifted by half the FSR with respect to each other) to be simultaneously closer to their respective resonance. In addition, we also provide in Supplementary Section II analytical estimates of the limits of the P2 SSB existence range which are found to be in excellent agreement with numerical results. We now cite these results in Section IV where we previously mentioned that δ_π had to be “within about the resonance linewidth” (and have removed the latter statement).

The Reviewer also inquires about the role of the ellipticity of the pump. Because of the half-FSR shift between the resonances of the x and y modes, any component of the driving field that is not polarized along the x mode is essentially in anti-resonance and does not get coupled into the resonator. Accordingly, the pump ellipticity simply affects the effective driving power, and any misalignment of the driving polarization with that of the x mode can always be compensated by driving the resonator stronger. This is now discussed in the Methods, below Eqs. (A.5)–(A.6).

b. “Over multiple runs, we have accumulated a sequence of 2.4 million individual events” Why did it require multiple runs to record the events instead of a single continuous measurement? How long in total does it take to record 2.4 million events? Did the authors have to readjust some parts of the system in between runs?

The need to collect data over multiple runs is related to limits in the triggering of our oscilloscope. The data is captured using the so-called segmented memory mode, which allows to capture a subset of the output signal at specified interval, without filling up the memory with all the intermediate data. There is however a limit on the maximum number of such “segments” that can be handled per run and the capture of 2.4 million events requires 2 to 4 runs (depending on sampling rate etc). In practice, we control the oscilloscope with a computer to run

the scope and transfer the data in quick succession automatically. The data transfer is the slowest part of the process and the overall acquisition of 2.4 million events takes a few seconds. No parameters are adjusted as we proceed. We have now clarified this aspect in our revised manuscript in the last paragraph of Section V.

c. “In fact, we observe this behavior as soon as the y -mode phase defect is sufficiently close to π (to within about the resonance linewidth).”

From the sentence above, can the condition “within about the resonance linewidth” be generalized for other systems? If not, could you elaborate on what needs to be considered?

Although the title of our manuscript refers to generic dissipative systems, we are more specifically focusing on resonator systems with driving where it is always possible to define a resonance linewidth. This may be different for other systems, such as bosonic systems (see also our response to point 4 of Reviewer 3). See also our response to point 2a above where we point out new results that show that the system can be detuned from an exact π offset by quite a bit more than the resonance linewidth as the driving power gets larger.

d. “Note that the above analysis assumes a π phase defect that is purely localized.” Considering the sentence above, mentioning the phase defect-to-cavity length ratio in the experiment would be helpful. Also, at what ratio will the y -mode intensity halve? Please answer if this is something trivial to extract. Although these are specific to the demonstrated experiment, this may assist audiences’ initial attempt to choose cavity length and phase defect component (mode converter or FPC).

In our experiment, the fiber polarization controller uses a bit less than one meter of fiber. So for our shorter resonator, the π phase defect is distributed over about 10 % (or a bit less) of the overall resonator length. We have added that information in our revised manuscript near the end of the Methods.

It is not trivial to predict for which defect-to-cavity length ratio the y -mode would halve. The equations describing such situations become complicated enough that they can only be solved numerically and the answer would depend on a number of parameters. However, the effect is quite robust. For some parameters, we have verified that the alternating P2 dynamics can persist even with the phase defect distributed over more than half the resonator length. Also, it is important to note that distributing the phase defect essentially amounts to reducing the parametric gain that generates the y -mode, which can always be compensated by driving the resonator harder. We have also added these details at the end of the Methods.

Minor comments:

1) Fig. 3 y -label: Intensity [a.u.] -> Intensity [a.u.]

This has been fixed.

2) Regarding SSB, I noticed a recent work [Moroney, Niall, et al. “A Kerr polarization controller.” Nature Communications 13.1 (2022): 398.]. Exposing the audience to a different take (exploiting the asymmetry for deterministic polarization control) on the polarization mode SSB would be beneficial. *Only if the authors agree that it is relevant to include it in the manuscript.

We thank the Reviewer for this suggestion and have added this reference, now cited in the Introduction as ref. [28].

Response to Reviewer #3

The paper presents a novel exploration into the fascinating interplay between topology, nonlinearity, and spontaneous symmetry-breaking (SSB) in the realm of driven Kerr resonators. The authors' attempt to demonstrate this complex relationship is commendable and highlights a significant step forward in the research in this field.

1. The authors have constructed a robust theoretical framework, pioneering the concept of a Möbius topology and linking it with SSB phenomena. However, some questions remain: The authors should comment on the effectiveness of their model at negative detuning, as the fitting to the experimental data becomes less accurate.

We believe the Reviewer is referring here to the fit between measurements and numerical predictions shown in Fig. 3(d), where the data at negative detuning indeed shows a slight departure from the numerics. We believe this small discrepancy can be attributed to uncertainties in the experimental parameters, in particular the detuning, the nonlinearity coefficient, and the cross-polarization coefficient B . We note that the latter typically departs from the idealized value of 2 because the polarization is actually evolving along the fiber in the experiment, but it is not trivial to assess that effect exactly. We now briefly comment on these discrepancy in our revised manuscript (end of Section IV).

2. The authors should provide more experimental evidence of the symmetry protection and stability. The authors refer to 30min stability limited only by the locking feedback loop, which I argue is also part of the system. I understand that this is not inherent to the SSB in the ring resonator, but more results, or at least arguments on the thermal and polarization stability in the fiber loop should be given.

We have added a new Supplementary Section V showing measurements of P2 domain walls and P2 bright symmetry-broken cavity solitons over periods of 30 to 60 minutes. It is referred to as at the end of Section VI of the main Article. We hope this satisfies the Reviewer's concerns.

We note that, after a period of warm up, thermal and polarization stability are not a concern and measurements can be taken over and over with no further adjustment. Our cavity detuning locking mechanism is then only limited by strong external perturbations, like somebody slamming the lab door.

3. The presented evidence, compellingly supports the existence of P2 domain walls — a noteworthy finding attributed to the Möbius topology. However, the connection between this unique topological feature and the emergence of these domain walls requires further clarification.

The P2 domain walls separate domains realizing the two different symmetry-broken states of the system. The way we generate them in our system is by ramping up the detuning through the threshold of spontaneous symmetry breaking. As doing so, different parts of the nanosecond driving pulses circulating in the resonator will break they symmetry in different directons, leading to the initiation of a random combinations of domains. We then lock the detuning at a set value and observe their stationary state over time. We have now added some clarifications regarding this point in Section VI, at the end of the paragraph where P2 domain walls are discussed.

4. The authors indicate the implications of their work might extend beyond photonics into other bosonic systems. While this proposition is intriguing, it would be beneficial to provide explicit examples of practical applications that could potentially be influenced by these findings. Doing so would underline the novelty and significance of their research. The work present great fundamental physics investigation, perhaps more emphasis on the applications is appreciated.

While applications of our study to bosonic systems are out of scope of the present work, we have added in the Discussion that our results directly apply to a broad range of two-mode bosonic settings, including two-component atomic gases and cold bosonic atoms trapped in a double well. More examples can be found, e.g., in Section VIII of arXiv:2304.05865 which discusses periodically-modulated two-mode bosonic systems, in quantum gases in particular. This arXiv reference is an extended version of former ref. [40], which now replaces it as ref. [41]. Another example is provided by Dogra et al [Science **366**, 1496 (2019)] who study the interplay between dissipative and unitary processes in spinor BECs. We now cite this reference [80] in the Discussion.

5. An interesting claim made by the authors is the extension of the robustness associated with the Möbius topology to complex inhomogeneous structures. This assertion could benefit from more supportive evidence, be it empirical or theoretical.

We believe the Reviewer refers to our statement at the end of Section VI. We note that the “complex inhomogeneous structures” we are referring to here are simply the ones we have reported in that section (P2 domain walls, P2 symmetry broken bright cavity solitons, and P2 asymmetric breathers). We do not mean anything else. Therefore the evidence is actually already presented in that section.

In summary, the manuscript provides a considerable and innovative contribution to its field and is well suited for publication in Nature Communications. I recommend acceptance, provided the authors address the queries and suggestions raised in their revised submission.

REVIEWERS' COMMENTS

Reviewer #1 (Remarks to the Author):

The authors provided convincing answers to my questions and to the comments of the other reviewers. the paper can be published as is.

Reviewer #3 (Remarks to the Author):

I am satisfied with the answers provided by the authors, I can recommend the paper for publication.